# Single-Cell RNA Sequencing of Baseline Immune Profiles After Third Vaccination Associated with Subsequent SARS-CoV-2 Infection in Naïve Individuals

**DOI:** 10.3390/ijms26083494

**Published:** 2025-04-08

**Authors:** Hyunhye Kang, Junseong Park, Hyunjoo Bae, Yeun-Jun Chung, Eun-Jee Oh

**Affiliations:** 1Department of Laboratory Medicine, Seoul St. Mary’s Hospital, College of Medicine, The Catholic University of Korea, Seoul 06591, Republic of Korea; azuresky@hanmail.net; 2Department of Laboratory Medicine, Shinwon Medical Foundation, Gwangmyeong-si 14316, Republic of Korea; 3Cancer Evolution Research Center, College of Medicine, The Catholic University of Korea, Seoul 06591, Republic of Korea; j.p@catholic.ac.kr; 4Precision Medicine Research Center, College of Medicine, The Catholic University of Korea, Seoul 06591, Republic of Korea; 5Department of Medical Sciences, Graduate School, The Catholic University of Korea, Seoul 06591, Republic of Korea; jaydcom8673@gmail.com; 6Department of Microbiology, College of Medicine, The Catholic University of Korea, Seoul 06591, Republic of Korea; 7Research and Development Institute for In Vitro Diagnostic Medical Devices, The Catholic University of Korea, Seoul 06591, Republic of Korea

**Keywords:** SARS-CoV-2, single-cell RNA sequencing, NK, interferon, vaccine, innate immunity, memory B cell

## Abstract

Even though vaccines protected many from infection, not all were protected, and vaccinated individuals displayed a wide range of clinical outcomes, from complete protection against infection to multiple breakthrough infections. This study aimed to identify baseline differences following identical ChAdOx1/ChAdOx1/BNT162b2 in infection-free and breakthrough-infected individuals to find molecular signatures linked to enhanced SARS-CoV-2 protection. Samples from a previous longitudinal study were analyzed, classifying subjects as ‘Protected’ or ‘Infected’ based on infection status over two years. SARS-CoV-2–specific immunological assays and single-cell RNA sequencing evaluated baseline differences. Although humoral response measurements showed no significant difference, enhanced cellular responses via enzyme-linked immunospot assays were observed in the Protected group. Differentially expressed genes and pathway analysis of T/NK subsets showed the Infected group had reduced inflammation and interferon responses. The Infected group also displayed downregulated interaction with CD4+ T cells. B subset analysis revealed more memory B cells in the Infected group, accompanied by downregulation of immune regulatory genes and upregulation of the small ubiquitin-related modifier pathway. Our findings revealed differential molecular signatures in the baseline immune subsets of vaccinated individuals with prolonged protection and breakthrough infection. Reduced immune regulation and altered cell interactions may contribute to breakthrough infection, providing insights for future vaccine development and targeted protective strategies.

## 1. Introduction

Numerous studies have underscored the crucial role of vaccination in mitigating severe acute respiratory syndrome coronavirus 2 (SARS-CoV-2) infections [1,2]. However, even though vaccines protected many from infection, not all were protected, and vaccinated individuals displayed a wide range of clinical outcomes, from complete protection against infection to multiple breakthrough infections [3,4,5,6].

To elucidate the heterogeneity of host immune responses following vaccination and infection, researchers have increasingly turned to single-cell RNA sequencing (scRNA-seq), as highlighted in previous studies [7,8,9]. Early in the pandemic, researchers focused on the transcriptomic differences between COVID-19 patients and healthy controls. Wilk et al. demonstrated defective immune defense mechanisms, including the depletion of innate immune subsets in infected individuals, with these defects being more pronounced in severely symptomatic patients, compared to both healthy controls and patients with moderate symptoms [10]. Subsequent studies have explored molecular-level immune response defects in severely symptomatic patients and reported the downregulation of human leukocyte antigen (HLA) class II-related genes and defective antigen processing pathways in COVID-19 patients [11,12]. Later studies revealed transcriptomic changes following vaccination. Sequential immune reactions following BNT162b2 (Pfizer–BioNTech, hereafter referred to as BNT) vaccination were investigated [13]. Additionally, comparative analyses explored the immune responses between vaccinated and naturally infected individuals [14], as well as the immunogenicity of ChAdOx1 (AstraZeneca, hereafter referred to as ChAd)/AZD1222 and mRNA-1273 vaccines in patients with autoimmune diseases [15]. Numerous topics have been addressed in the extant literature; however, to the best of our knowledge, limited attention has been given to identifying the vaccine-induced molecular signature that correlates with later infection outcomes.

Given that not all individuals respond equally to vaccination and that instances of vaccine failure exist [16], investigating the heterogeneous clinical outcomes following identical vaccination regimens is crucial for future vaccine development. For example, single-cell transcriptomic analyses have been conducted to study vaccine non-responders in the context of hepatitis B virus (HBV) vaccination [17]. However, in the case of SARS-CoV-2, there remains a significant gap in understanding the single-cell-level immune response heterogeneity after vaccination that contributes to the varying degrees of protection against infection. We hypothesized that differences in baseline molecular characteristics may account for the heterogeneous infection outcomes among a relatively homogenous cohort of the same vaccine type. To this end, we employed scRNA-seq, as it has been widely and effectively used to decipher transcriptomic differences between groups.

We conducted a retrospective analysis of baseline molecular responses in immune subsets following identical vaccine schedules in individuals with prolonged protection and breakthrough infection. Our aim was to compare the vaccine-induced immune response at a high-resolution single-cell level and to identify differential responses that could be relevant to subsequent SARS-CoV-2 infection.

## 2. Results

### 2.1. Subjects Characteristics

The Protected group included females aged 27–53 years old, and the Infected group included females aged 37–59 years old (median 49, interquartile range 45.0–51.5). All subjects completed two doses of the ChAdOx1 nCoV-10 primary series and one booster dose of BNT162b2. The third dose was administered between November and December 2021. The duration from vaccination to sampling ranged from 145 to 166 days in the Protected group and from 96 to 176 days in the Infected group. Most subjects were healthy, without underlying diseases, except for two individuals from the Infected group. One subject reported underlying well-controlled hypertension, and the other reported type 2 diabetes mellitus.

Among the 12 subjects in the Infected group, four individuals underwent scRNA seq. Their ages ranged from 46 to 49 years old. The duration from the third dose to sampling ranged from 96 to 175 days. The characteristics of subjects are summarized in Table 1.

### 2.2. SARS-CoV-2–Specific Surrogate Humoral and Cellular Immune Responses

To evaluate whether there was a significant difference in SARS-CoV-2-specific humoral immune response at baseline, we compared the surrogate virus neutralization test (sVNT) results between the Protected and the Infected groups. The median percent inhibition measured by sVNT (Figure 1A) was not significantly different between the Protected and the Infected groups (*p* = 0.2963).

We used an enzyme-linked immunospot (ELISpot) assay as a surrogate indicator of cellular immune response. When stimulated by SARS-CoV-2 peptides, a significantly greater magnitude of interferon gamma (IFN-γ) was produced in the Protected group compared to the Infected group (*p* = 0.0208) (Figure 1B). Overall, we observed a significantly enhanced cellular immune response in the Protected group, which persisted 4–6 months after administration of the third dose, while the humoral immune response exhibited inappreciable differences.

### 2.3. Comparison of Gene Expression and Distribution of Total Immune Cells

scRNA-seq was employed to examine the transcriptomic characteristics induced by the vaccine. The scRNA-seq data statistics are summarized in Appendix A. Following the acquisition of data from individual samples, subsequent analyses were conducted on pooled samples according to the group definition. After quality control, a total of 30,208 cells were obtained, with 11,494 from the Protected group and 18,714 from the Infected group. UMAP visualization of total single cells showed similar gene expression patterns between the Protected and the Infected groups (Figure 2A). Based on the distinct expression levels of canonical marker genes (Appendix A), we identified five immune cell subtypes: “T/NK”, “Myeloid”, “B”, “Plasma”, and “Erythrocyte” (Figure 2B,C). The distribution of these clusters and the proportions of these five immune cell subtypes did not differ between the Protected and the Infected groups (Figure 2D). We then conducted further clustering for three cell types: T/NK cells, myeloid cells, and B cells.

### 2.4. Comparison of Single-Cell Transcriptional Characteristics of T/NK and Myeloid Subsets

Through additional clustering, we identified six T cell subpopulations (CD4+ helper T, CD4+ naïve T, TCR-low T, CD8+ effector T, CD8+ naïve T, Treg) and one NK cell population from the “T/NK” cluster (Figure 3A,B). The proportion of these seven subpopulations showed no significant difference between the Protected and the Infected (Figure 3C). However, the analysis of gene expression in each subcluster revealed that not only were IFN-γ-related upregulated but several other important genes involved in signaling pathways were also upregulated in NK cells in both groups, with a greater degree in the Protected group. The Protected group also showed greater expression of the *TNFAIP3* gene in all T/NK subsets, which may play a role in cytokine-mediated immune and inflammatory responses (Figure 3D). While NK cells are traditionally associated with recognizing stress signals and abnormal host cells, they also express receptors that can recognize pathogen-associated molecular patterns (PAMPs), which are conserved molecular motifs found in pathogens. Key receptors involved in inflammatory pathways and PAMP recognition in NK cells include Toll-like receptors (TLRs), NOD-like receptors (NLRs), RIG-I-like receptors (RLRs), and c-type lectin-like receptors (CLRs). Among these receptors, our analysis revealed that *IFIH1* showed the highest expression levels in the NK cell population (Figure 3E).

Next, enrichment analysis using gene sets derived from canonical pathway showed that the Infected group had downregulated inflammation and interferon responses across the T/NK subsets (Figure 3F). In particular, pathways involved in the IFN-γ response and host–pathogen interactions related to SARS-CoV-2-induced IFN responses were more enriched in the NK cells of the Protected group.

Myeloid cells were divided into four subpopulations: CD14+ monocytes, CD16+ monocytes, intermediate monocytes, and dendritic cells (Figure 4A,B). The Infected group exhibited a higher proportion of CD14+ monocytes and a lower proportion of CD16+ monocytes compared to the Protected group (*p* = 0.003, Figure 4C). However, considering the highly variable numbers of monocytes within the Infected group, we cannot conclude that this trend is a meaningful observation. Although the percentages of immune cells did not show remarkable differences, cell–cell interaction analysis revealed that cellular interactions through CD16 decreased in the Infected group, while interactions through dendritic cells increased (Figure 4D,E). In particular, the analysis of interactions mediated by MHC molecules showed downregulated interactions via certain MHC class II molecules with CD4+ naïve and CD4+ helper T cells (Figure 4F). Specifically, the interactions via HLA-DQA2 between CD14+ monocytes and CD4+ T cells were downregulated, similar to interactions via HLA-DQB between dendritic cells and CD4+ T cells.

### 2.5. Comparison of Single-Cell Transcriptional Characteristics of B Cells

Further analysis of the B cells revealed two distinct subpopulations: naïve and memory B cells (Figure 5A,B). The proportion of memory B cells was significantly higher in the Infected group compared to the Protected group (*p* < 0.001, Figure 5C). Subsequently, we examined the molecular characteristics of memory B cells by comparing the DEGs between the Protected and the Infected subjects. Remarkably, the Infected subjects showed downregulation of several genes that are crucial for immune cell regulation, including *LY9*, *CD83*, and *CD69*, which are associated with lymphocyte differentiation and activation, as well as *CCR7*, a key chemokine receptor, and *NFKBID*, a gene encoding an NF-κB inhibitor (Figure 5D). Collectively, these findings suggest that the Infected subjects have a reduced capacity for lymphocyte regulation at baseline, which may contribute to their increased susceptibility when encountering the target pathogen. Pathway analysis revealed that inflammatory pathways were downregulated in the Infected group compared to the Protected group (Figure 5E), while the small ubiquitin-related modifier (SUMO) pathway was more enriched in the Infected group.

## 3. Discussion

This study aimed to compare immune responses following vaccination at the single-cell level between subsequent infection-protected individuals and breakthrough-infected individuals, with a focus on identifying molecular signatures that may impact infection outcomes. The objective of this study was to assess vaccine-induced immune response, which potentially confers long-term protection when we exclude factors claimed to be associated with an impact on vaccine-induced immunity. We included samples matched for sex, vaccine schedules, and number of administered doses. The infection-protected individuals in our cohort were exclusively females [5]. Previous research has shown that sex is a significant correlate of breakthrough infection, although contradictory observations have been made regarding which sex poses a vulnerability [18,19]. There was no significant difference in vaccine schedule, vaccine type, or number of vaccine doses across sexes in our cohort [5]. However, Yamal et al. argued that sex is a significant factor for breakthrough infection after adjusting for age, race, vaccine type, and boosters [19]. Previous studies have posited that adenovirus DNA vector vaccines may place a greater number of individuals at increased risk of breakthrough infection compared to mRNA-based vaccines [19,20]. A previous nationwide retrospective cohort study in South Korea demonstrated the beneficial impact of a BNT booster, indicating that the booster could mitigate breakthrough infections in ChAd-primed individuals [20]. Booster vaccination is associated with a lower breakthrough infection rate [21] and a greater immune response [3,5]. We evaluated samples following the administration of the third dose, as previous research has demonstrated that the third antigenic stimulus elicits the most potent immunogenic response [22,23], with the immune response persisting robustly for over one year post-administration [5]. To focus specifically on vaccine-induced responses, we excluded individuals with prior infection and conducted a comparative analysis of the infection-naive state based on retrospectively defined samples.

We observed significantly increased production of IFN-γ in the ELISpot assay for the Protected group and postulated that this enhanced cellular response may confer greater protection. To analyze the cellular response of each immune subset, we performed scRNA-seq. We first analyzed the proportion of T and NK cells and found no significant differences. Our findings are consistent with a previous study that demonstrated no significant differences in the number and proportion of NK cells after vaccination [24]. However, the response of antigen-specific NK cells was not examined in this study, despite several significant previous studies reporting notable changes in antigen-specific T cells following vaccination [13,25,26]. Prior research has indicated that antigen-specific memory cells are formed in NK cells through vaccines or infections [27,28]. An investigation of changes in antigen-specific NK cells would be a valuable additional study.

Differences in IFN-γ pathway activity in NK cells between the Protected and the Infected groups can arise from a variety of mechanisms, including changes in signaling pathways, cellular microenvironments, and epigenetic or transcriptional regulation. Specifically, the activation of pattern recognition receptors (PRRs), stress signals, pro-inflammatory cytokines such as IL-12, IL-18, and type I IFNs, transcription factors such as STAT1, STAT4, and NF-κB, as well as metabolic reprogramming can occur during infection, further activating the IFN-γ pathway. These factors are regulated by diverse mechanisms, including post-translational modification, subcellular localization, endocytosis, and epigenetic regulation, as well as their expression levels.

The cytotoxicity of NK cells varies in response to PRR activation through both direct and indirect mechanisms. PRR activation primes NK cells to respond more robustly, influences the signaling pathways they employ, and alters their interaction with target cells. Acute PRR activation leads to robust but transient cytotoxic responses. Chronic or excessive PRR activation can lead to NK cell exhaustion, reducing cytotoxicity. NK cell cytotoxicity plays several essential roles during infections, contributing to the early control of pathogen spread, the amplification of immune responses, and tissue homeostasis.

Notably, we observed differences in the DEGs and enriched pathways of the immune subsets between the Protected and the Infected groups. A general decrease in the expression of the inflammatory pathways was observed in the Infected group. Importantly, the Protected group exhibited a greater enrichment of IFN-γ-related pathways in NK cells compared to the Infected group. NK cells are critical effectors of the innate immune response and interact with other immune subsets, playing a crucial role in mediating the cytotoxic elimination of tumors and virus-infected cells by secreting IFN-γ [29]. In the context of vaccine-induced immune responses, NK cells can not only induce favorable short-term outcomes through innate immune responses but also boost vaccine-induced adaptive immunity. There is growing evidence supporting the adaptive-like functionality of NK cells after vaccination [29,30,31,32,33,34]. A recent single-cell multi-omics analysis identified differential expression of NK cell markers between vaccinated individuals and infection-free healthy controls, demonstrating that vaccinated individuals exhibit increased activation of NK cells [35]. NK cells can provide increased responsiveness to target pathogens for months through trained innate immunity [31]. Severa et al. also argued that SARS-CoV-2 vaccines induced IFN-γ–mediated innate immune response, which has a significant impact on adaptive immune responses and correlates with the humoral response [34]. Our findings suggest the hypothesis that the enriched IFN-γ–related pathways in NK cells may be associated with the increased IFN-γ production observed in the Protected group, requiring validation through further in vitro studies.

Our findings also indicated that several cell–cell interactions involving HLA class II molecules were downregulated in the Infected group. This suggests that antigen presentation to CD4+ T cells was less effective in the Infected group. Previous scRNA-seq studies have demonstrated that patients exhibiting impaired antigen presentation by dendritic cells and/or CD14+ monocytes are more susceptible to severe infections [10,11,12]. Consistent with previous observations, our findings suggest that defective antigen presentation could be associated with reduced vaccine effectiveness and may contribute to a higher risk of breakthrough infections.

The development of memory B cells is crucial to ensure optimal vaccine effectiveness. Unlike waning antibodies after every vaccination, memory B cells persist and induce the production of large amounts of neutralizing antibodies upon viral entry [36]. These neutralizing antibodies play an important role in protection against SARS-CoV-2 infection as well, by blocking the interaction between the viral spike protein and the angiotensin converting enzyme 2 (ACE-2) receptor [37]. Although vaccination typically induces robust memory B cell populations, our analysis revealed a paradoxical increase in memory B cell frequency in the Infected group, despite a diminished capacity to efficiently activate lymphocytes. This apparent contradiction reflects qualitative differences in memory B cell subsets and the viral subversion of post-translational regulatory pathways. The significant upregulation of the SUMO pathway in the Infected group is a noteworthy finding. SUMOylation, a post-translational modification, plays a vital role in enhancing host defense mechanisms against pathogens by altering type 1 IFNs, thereby regulating the innate immunity [38,39,40]. However, some viruses have been reported to manipulate the SUMO pathway, which can impair innate immunity of the host and contribute to successful viral infections [41]. Existing literature reports various mechanisms of modulation by various pathogens in the SUMO pathway. Li et al. argued that SUMOylation of nucleocapsid protein was found to help its homo-oligomerization, which is a critical event for SARS-CoV infectivity [42]. Although there has not been a conclusive observation whether the nucleocapsid protein of SARS-CoV-2 is SUMOylated, some studies have suggested the possibility based on a high degree of similarity between nucleocapsid protein sequences among coronaviruses. In addition, activated SUMOylation is hypothesized to suppress MHC class I antigen presentation, potentially conferring immune evasion in cancer [43]. Further research is required to determine whether this mechanism is applicable in SARS-CoV-2 infection.

The present study has several limitations that must be acknowledged. First, several factors restrict the generalizability of this study; we analyzed a limited number of subjects, all of whom were female, and there was an imbalance in age distribution between the groups. Because the number of breakthrough infections in our cohort was relatively small, this study may be underpowered to detect subtle transcriptomic differences, which requires future studies, with larger and independent populations. Since the impact of sex on immune response post-vaccination remains inconclusive, and minors and the elderly were not included in this study, we believe their influence on the current findings would be minimal, if any. As our study involved participants who had received three doses, including a booster known to enhance vaccine efficacy, the conclusions drawn may differ from those involving individuals with fewer vaccinations. Although the sample size in the current study is small, the molecular characteristics identified provide insights that may inform future studies aimed at understanding vaccine responses. In addition, the participants’ mild symptom presentation confines the applicability of results to individuals with less severe infections. It is worth noting that patients experiencing more severe symptoms may potentially display distinct immunological responses. Second, we evaluated samples obtained at only one time point, which was 4–6 months after the third dose. Third, we were unable to analyze the immune response before vaccination and after each vaccine dose, which prevented us from clarifying the impact of an individual’s pre-vaccination immune status and that of each vaccine regimen on the immune response. Finally, neutralizing antibodies were assessed using a commercial ACE2 binding inhibition assay kit, rather than a pseudovirus neutralization assay. The latter methodology would yield more accurate estimations of neutralizing antibody titers.

## 4. Materials and Methods

### 4.1. Study Design and Subjects

This study is part of a follow-up series to our previous longitudinal cohort study, in which we recruited 78 healthcare providers from Seoul St. Mary’s Hospital to investigate longitudinal immune response following vaccination and SARS-CoV-2 infection [5]. Briefly, infection-naïve participants were enrolled in March 2021 and followed longitudinally until June 2023 to determine their immune response and infection status. Based on self-reported infection history as of June 2023, participants were classified as either Protected or Infected. To ensure accurate classification, all participants in the Protected group were confirmed to be naïve through negative anti-nucleocapsid antibody (anti-N) testing (Roche Diagnostics, Basel, Switzerland). Given this group definition, we conducted a retrospective investigation of samples obtained from February to May 2022, corresponding to 4–6 months after the administration of the third vaccination dose. To exclude the effect of prior SARS-CoV-2 infection, we included only participants who were confirmed to be uninfected with SARS-CoV-2 by the time of sampling and who were nonreactive to anti-N testing. The three subjects in naïve group were all females and vaccinated with ChAdOx1/ChAdOx1/BNT162b2. Twelve subjects in the Infected group shared the same characteristics and were included in the current study. Of the 12 samples, four were available for scRNA-seq. The scheme illustrating the subject definition and selection criteria is presented in Figure 6.

This study was approved by the Institutional Review Board of Seoul St. Mary’s Hospital (KC23TISI0183). All the participants provided written informed consent.

### 4.2. Assays for SARS-CoV-2-Specific Humoral and Cellular Immune Responses

Humoral immune responses were measured as previously described [5]. Briefly, the neutralizing activity against wild-type SARS-CoV-2 was evaluated using a commercially available sVNT (GenScript, Piscataway, NJ, USA) according to the manufacturer’s instructions. Samples with positive and negative controls were diluted in tubes with horseradish peroxide-conjugated recombinant receptor-binding domain of the viral and human angiotensin converting enzyme 2 at a volume ratio of 1:1. After incubation and washing, the reaction was read at 450 nm wavelength using an enzyme-linked immunosorbent assay microplate reader. The percent inhibition was calculated by subtracting the optical density ratio of the sample and the negative control from 100% (%inhibition = (1 − OD value of sample/OD value of negative control) × 100%). The manufacturer suggested a cutoff of 30% was applied to determine positive neutralizing activity.

We also evaluated the in vitro SARS-CoV-2–specific cellular immune response using ELISpot with the BD Human IFN-γ ELISpot kit (BD Biosciences, Franklin Lakes, NJ, USA), as previously described [44]. Briefly, peripheral blood mononuclear cells (PBMCs) were stimulated with PepTivator SARS-CoV-2 wild-type S peptide pools (Miltenyi Biotec, Bergisch Gladbach, Germany) comprising 15 mer sequences with an 11 amino acid overlap. PBMCs were also stimulated with phorbol 12-myristate 13-acetate/ionomycin as a positive control, and RPMI medium as a negative control. The IFN-γ-forming spots were read using an AID iSpot ELISpot fluorospot reader, and the average spot count per 2.5 × 10^5^ PBMCs was calculated from duplicate measurements after subtracting the negative control response.

### 4.3. Preparation of Single-Cell Suspensions for scRNA-Seq

After blood collection, PBMCs were immediately isolated using Ficoll-Paque Plus (GE Healthcare, Chicago, IL, USA) and then cryopreserved. The cell stocks were thawed at 37 °C in 10% fetal bovine serum/Dulbecco’s Modified Eagle Medium and washed twice with cold Ca^2+^- and Mg^2+^-free 0.04% bovine serum albumin/phosphate-buffered saline (BSA/PBS) at 300 g for 5 min at 4 °C. The cells were gently resuspended in BSA/PBS and counted using a LUNA-FX7 Automated Fluorescence Cell Counter (Logos Biosystems, Anyang-si, Republic of Korea) with Acridine Orange/Propidium Iodide staining. Cell viability was further assessed using the Dead Cell Removal Kit (Miltenyi Biotech) and MS columns (Miltenyi Biotech), according to the manufacturer’s instructions. For multiplexing, each sample was labeled with an antibody-polyadenylated DNA barcode for human cells (BD Biosciences). Briefly, the cells were stained with the multiplexing antibody for 20 min at room temperature, followed by three washes with staining buffer (BD Biosciences). After the final wash, the samples were gently resuspended in cold sample buffer (BD Biosciences), counted, and pooled.

### 4.4. Single-Cell RNA Sequencing

Single-cell capture was performed using a BD Rhapsody Express instrument in accordance with the manufacturer’s instructions. Briefly, pooled cells from each sample resuspended in cold sample buffer were loaded into a BD Rhapsody cartridge (BD Biosciences). Following cell separation, cell barcode–magnetic beads were loaded into the cartridge. Subsequently, the cells were lysed, and mRNA capture beads were retrieved. cDNA synthesis and Exo-nuclease I treatment were performed on the mRNA capture beads using a BD Rhapsody cDNA kit (BD Biosciences).

scRNA-seq libraries were generated using the BD Rhapsody WTA amplification kit (BD Biosciences) according to the manufacturer’s protocol. Purified WTA and sample tag libraries were quantified by qPCR. Libraries were pooled and sequenced on the NovaSeq platform (Illumina, San Diego, CA, USA) to produce 100 bp paired-end reads. The resulting FASTQ files were analyzed using the BD pipeline on the SevenBridges platform. The statistics of the raw scRNA-seq data are summarized in Appendix A.

Doublets were predicted and excluded using DoubletFinder [45], and low-quality cells with <200 genes or >30% mitochondrial genes were filtered (Appendix A). All individual datasets were then integrated with batch correction using the Harmony algorithm [46]. Subsequent analyses, including normalization (SCTransform), unsupervised clustering, UMAP dimensionality reduction, and differentially expressed gene (DEG) analyses, were conducted using the R package Seurat (v5.0.1) [47] to identify, characterize, and visualize clusters, as previously described [48]. The cell types were manually annotated by comparing the canonical marker genes and differentially expressed genes (DEGs) for each cluster. Enrichment analyses were performed using the R package UCell (version 2.10.1) [49], using gene sets derived from the c2 canonical pathways of MSigDB. Cell–cell interactions were analyzed using the R package CellChat (version 2.1.2) [50].

### 4.5. Statistical Analysis

Categorical data are presented as counts and percentages, whereas continuous variables are presented as medians and interquartile ranges. The Mann–Whitney-U test was implemented to assess serological assay values, and the Chi-square test was applied to calculate the proportion of cells between groups. The correction for multiple comparisons was not available in all cases, and *p* values of less than 0.05 were considered statistically significant. The analysis and visualization of the data were performed using the Prism version 10.0.2 for Windows (San Diego, CA, USA), as well as R version 4.3.3.

## 5. Conclusions

Understanding the differential responses to vaccines of individuals who are protected from infection and those who experienced breakthrough infections is crucial for developing effective and targeted vaccination strategies. In this study, we followed healthy individuals for over two years after SARS-CoV-2 vaccination to identify infected and uninfected individuals and evaluated the differences in baseline immune signatures between them. In particular, this study highlights the potential implications for future vaccine development by comparing the molecular characteristics of individuals who were vaccinated but did not experience infection and those who did. The findings support the role of NK cells in preventing infection post-vaccination, suggesting that enhancing adaptive-like functionality in future vaccines could improve their effectiveness. Additionally, this study identifies defective antigen presentation as a factor in infection susceptibility, indicating the need for a vaccine that addresses this issue. The relationship between SUMOylation and SARS-CoV-2 suggests that controlling this mechanism could enhance vaccine efficacy. However, while this study provides valuable comparisons between groups, further experimental research is needed to clarify the causal relationships. Our findings may have important implications for developing more effective and targeted vaccination strategies.

## Figures and Tables

**Figure 1 ijms-26-03494-f001:**
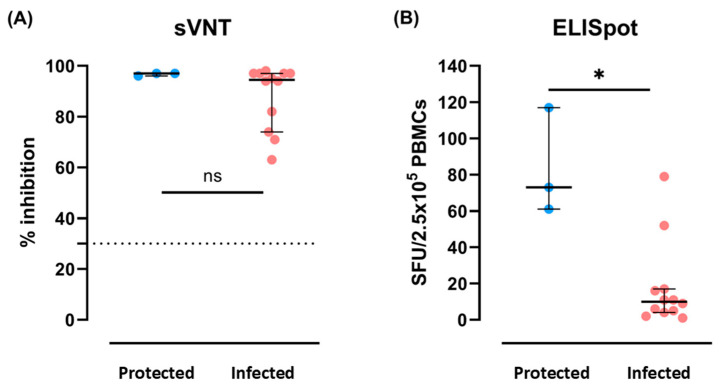
Comparison of baseline SARS-CoV-2-specific neutralization (**A**) and IFN-γ production (**B**) among study participants. Blue dots represent an individual’s assay results from the subjects in the Protected group, whereas red dots represent the results from the Infected group. Horizontal bars indicate the median, and error bars represent interquartile ranges. The horizontal dotted line indicates the suggested assay cut-off in A. *p* values were calculated using the Mann–Whitney U test, showing a non-significant difference in sVNT and significantly greater IFN-γ production in the Protected group (*p* = 0.0208). ns, not significant; * *p* < 0.05.

**Figure 2 ijms-26-03494-f002:**
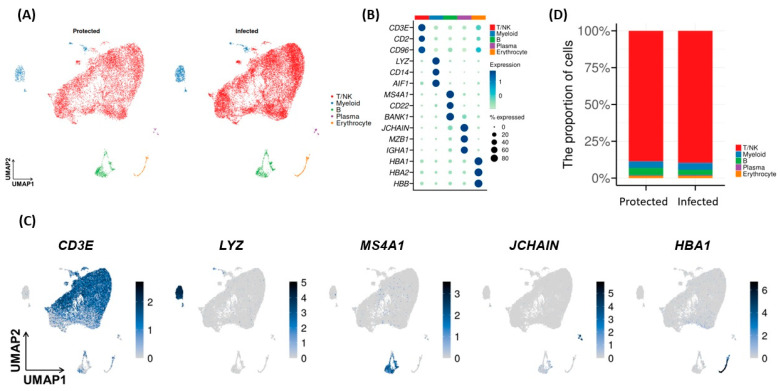
Clustering and cell type annotation of scRNA-seq data. (**A**) UMAP plots colored by clusters. (**B**,**C**) The cell types were manually annotated based on the expression of canonical marker genes. A dot plot shows the average expression of canonical cell type markers for each cluster (**B**), and UMAP plots show representative markers (**C**). (**D**) Bar plot showing the proportion of clusters according to groups (Chi-square test: *p* < 0.001).

**Figure 3 ijms-26-03494-f003:**
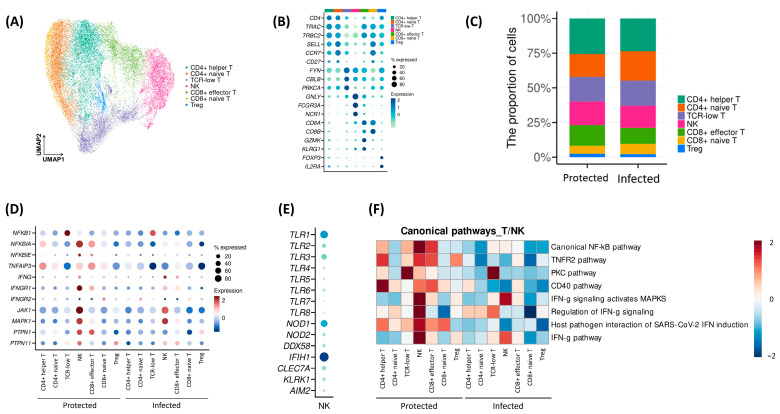
Subset analysis of the “T/NK” and “Myeloid” clusters. (**A**) UMAP plot of the “T/NK” cluster colored by subclusters. (**B**) Dot plot showing the average expression of representative markers for each “T/NK” subcluster. (**C**) Bar plots showing the proportion of “T/NK” subclusters according to groups (Chi-square test: *p* < 0.001). (**D**) Dot plot showing the average expression of several immune-associated genes for each group. (**E**) Dot plot showing the average expression of receptors recognizing pathogen-associated molecular patterns in NK population. (**F**) Heat map obtained from the enrichment analysis for gene sets associated with canonical pathways.

**Figure 4 ijms-26-03494-f004:**
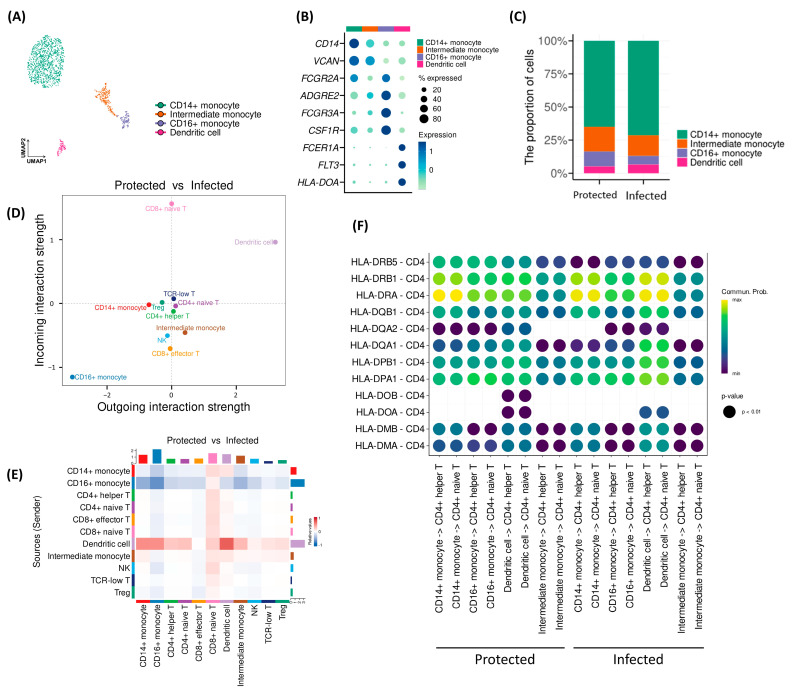
Subset analysis of the “T/NK” and “Myeloid” clusters. (**A**) UMAP plot of the “Myeloid” cluster colored by subclusters. (**B**) Dot plot showing the average expression of representative markers for each “Myeloid” subcluster. (**C**) Graphs showing the proportion of “Myeloid” subclusters according to groups (Chi-square test: *p* = 0.003). (**D**,**E**) Cell–cell interactions among “Myeloid” and “T/NK” subclusters were calculated and visualized as a scatter plot (**D**) and a heat map (**E**). (**F**) Cell-cell interactions from “Myeloid” subclusters to CD4+ and CD8+ T cells were evaluated, and signal intensities of each ligand–receptor pair were visualized as a bubble plot.

**Figure 5 ijms-26-03494-f005:**
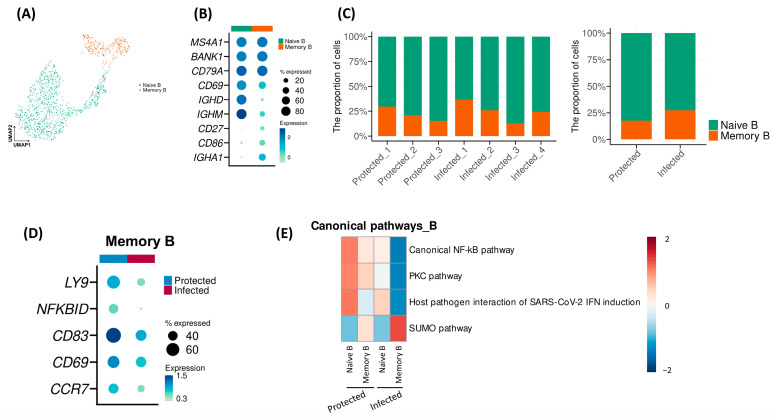
Subset analysis of the “B” clusters. (**A**) UMAP plot of the “B” cluster, colored by sub-clusters. (**B**) Dot plot showing the average expression of representative markers for each “B” subcluster. (**C**) Graphs showing the proportion of “B” subclusters of each sample (**left**) and according to groups ((**right**); Chi-square test: *p* < 0.001). (**D**) Dot plot showing the average expression of several immune-associated genes for each group. (**E**) Heat map obtained from the enrichment analysis for gene sets associated with canonical pathways.

**Figure 6 ijms-26-03494-f006:**
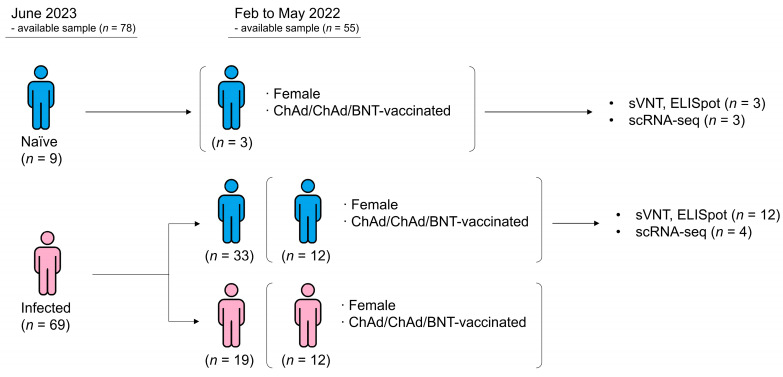
Schematic representation of subject definition and enrollment. The blue figure represents non-infected subjects, while the red figure represents infected subjects. Subjects who had not experienced SARS-CoV-2 infection are categorized as “Protected”, while subjects who had experienced at least one breakthrough infection until June 2023 are defined as “Infected.” The current study used this group definition and analyzed samples obtained from February to May 2022. Three patients in the Protected group underwent sVNT, ELISpot, and scRNA-seq, whereas 12 patients in the Infected group underwent all of the sVNT and ELISpot tests, but only four cases with sufficient sample volume underwent scRNA-seq. ChAd, ChAdOx1; BNT, BNT162b2; sVNT, surrogate virus neutralization test; ELISpot, enzyme-linked immunospot; scRNA-seq, single-cell RNA sequencing.

**Table 1 ijms-26-03494-t001:** The characteristics of enrolled participants.

Group	Age	Underlying Condition	Date of 3rd Dose Administration	Date of Sampling	3rd to Sampling (Days)	Date of COVID-19 Infection	Tests Conducted
Protected	27	none	16 November 2021	29 April 2022	164	NA	sVNT, ELISpot, scRNA-seq
Protected	53	none	17 December 2021	11 May 2022	145	NA	sVNT, ELISpot, scRNA-seq
Protected	53	none	17 November 2021	2 May 2022	166	NA	sVNT, ELISpot, scRNA-seq
Infected	47	none	15 November 2021	2 May 2022	168	October 2022 *	sVNT, ELISpot, scRNA-seq
Infected	49	none	12 November 2021	6 May 2022	175	1 January 2023	sVNT, ELISpot, scRNA-seq
Infected	46	none	11 November 2021	15 February 2022	96	16 March 2022, March 2023 *	sVNT, ELISpot, scRNA-seq
Infected	49	hypertension	15 November 2021	4 March 2022	109	16 March 2022, 16 December 2022	sVNT, ELISpot, scRNA-seq
Infected	51	none	16 November 2021	11 May 2022	176	October 2022 *	sVNT, ELISpot
Infected	44	none	17 November 2021	28 February 2022	103	10 March 2022, 7 February 2023	sVNT, ELISpot
Infected	56	none	10 November 2021	26 April 2022	167	October 2022 *	sVNT, ELISpot
Infected	59	none	19 November 2021	29 April 2022	161	October 2022 *	sVNT, ELISpot
Infected	50	DM	18 November 2021	2 May 2022	165	October 2022 *	sVNT, ELISpot
Infected	52	none	11 November 2021	15 February 2022	96	August 2022 *	sVNT, ELISpot
Infected	37	none	18 November 2021	3 May 2022	166	December 2022 *	sVNT, ELISpot
Infected	40	none	12 November 2021	6 May 2022	175	November 2022 *	sVNT, ELISpot

* Date not specified. Abbreviations: NA, not available; DM, diabetes mellitus; sVNT, surrogate virus neutralization test; ELISpot, enzyme-linked immunospot; scRNA-seq, single-cell RNA sequencing.

## Data Availability

Given the sensitive nature of patient data included in this study, the datasets generated and analyzed herein are not publicly accessible. The data presented in this paper are available upon request from the corresponding author.

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
