# Peer review of "Single-Cell RNA Sequencing of Baseline Immune Profiles After Third Vaccination Associated with Subsequent SARS-CoV-2 Infection in Naïve Individuals"

_ijms, 2025, doi:10.3390/ijms26083494_

Round 1
Reviewer 1 Report
Comments and Suggestions for Authors
Kang et al., present their findings on single-cell RNAseq of samples from ChAdOx1/ChAdOx1/BNT162b2 vaccinated people who were either protected from SARS-CoV-2 infection or experienced breakthrough infections in their manuscript entitled "Single-cell RNA Sequencing of Baseline Immune Profiles After Third Vaccination Predicts Subsequent SARS-CoV-2 Infection in Naive Individuals".
The manuscript is well written and provides relevant information. The authors find no significant differences in humoral response measurements but enhanced cellular responses in the protected group whereas the infected group exhibited reduced inflammation and interferon responses as well as downregulated interaction with T cells and increased memory B cells.
A general concern with the study design is that there is no confirmation that the "naive" group was exposed to SARS-CoV-2 and due to the vaccination protected from disease. I would also suggest to call the "naive" group "protected".
It would enhance the study if it would analyze female and male patients or look at other vaccines.
They found more memory B cells in the infected group - this needs to be discussed more deeply concerning the role of memory B cells in infection / vaccination.
Was there an analysis of which SARS-CoV-2 variant the patients were infected with and was the GenScript kit tested for its ability to neutralize other variants than wild-type SARS-CoV-2? A description of how long patients experienced infection related symptoms as well as a stratification of symptoms into mild-severe might be helpful.
There should be a paragraph in the discussion stating how these data are relevant for future vaccine development and targeted protective strategies as state in the abstract. Additionally, the title might have to be toned down if the manuscript does not clearly state which parameters can be used to predict infection breakthrough in vaccinated individuals.
Reviewer 2 Report
Comments and Suggestions for Authors
This manuscript presents a study investigating the baseline immune profiles of individuals following a third SARS-CoV-2 vaccination and their association with subsequent infection. The study utilizes single-cell RNA sequencing (scRNA-seq) to explore immune cell populations and gene expression differences between infection-naïve and breakthrough-infected individuals. The findings provide valuable insights into immune mechanisms that may contribute to vaccine-induced protection or susceptibility to breakthrough infection.
The study is well-structured and presents a compelling narrative supported by robust experimental methodologies. However, some areas require clarification and refinement to improve the manuscript's overall clarity, organization, and scientific rigor. Below, I provide general comments on key aspects of the manuscript, followed by specific comments with line numbers and figure-specific suggestions.
General comments:
- The manuscript provides a well-defined research question; however, the hypothesis should be more explicitly stated in the introduction. Clarifying the rationale behind using scRNA-seq to predict infection outcomes would strengthen the manuscript.
- While the results are presented systematically, some interpretations, particularly regarding the role of NK cells and the SUMO pathway, need further discussion to distinguish correlation from causation.
- The statistical methods used to compare immune responses between groups should be described in greater detail. Were corrections for multiple comparisons applied?
- The study includes a limited number of participants, all female, with an age imbalance between groups. The authors acknowledge this limitation, but additional discussion is needed on how this may impact generalizability.
- The figures are generally well-prepared, but some legends lack sufficient detail. For instance, a clearer description of the clustering approach in scRNA-seq would help non-specialist readers.
- Some figures (e.g., Figures 3 and 4) display numerous subpanels that could benefit from a more structured layout.
- The discussion effectively contextualizes findings but would benefit from a more direct comparison with previous scRNA-seq studies on vaccine responses.
- The role of the SUMO pathway in viral immune evasion is mentioned, but no experimental validation is provided. This should be framed as a hypothesis for future research rather than a definitive conclusion.
- Figure 1: The figure compares neutralization and IFN-γ responses between groups. The legend should specify whether the differences observed were statistically significant.
- Figure 2: The cell-type annotation process should be more clearly explained. How were marker genes selected?
- Figure 3: The multiple subpanels (A–L) make it difficult to track comparisons across groups. Consider grouping similar analyses together.
- Figure 4: The bar plot in panel C showing memory B cell proportions should include individual data points to better illustrate variability.
- Figure 5: The schematic representation is useful, but it would be helpful to indicate how samples were selected for scRNA-seq versus other assays.
